# Genomic diversity of diarrheagenic multidrug-resistant *Escherichia coli* across asymptomatic children and livestock in Nairobi, Kenya

Noah O. Okumu[1,2]*, John Juma[1], Samuel Oyola[1], Arshnee Moodley[1,3], Kennedy Mwangi[1], Gilbert Kibet[1], Linnet Ochieng[1], Julie Watson[4], Joseph J.N. Ngeranwa[2], Oliver Cumming[4], Elizabeth A.J. Cook[1], Dishon M. Muloi[1]

1 Health Department, International Livestock Research Institute, Nairobi, Kenya, 2 Department of Biochemistry, Biotechnology and Microbiology, Kenyatta University, Nairobi, Kenya, 3 Department of Veterinary and Animal Sciences, University of Copenhagen, Frederiksberg C, Denmark, 4 Department of Disease Control, London School of Hygiene and Tropical Medicine, London, United Kingdom

* nokmus02@gmail.com, N.Okumu@cgiar.org

## Abstract

Diarrheagenic *Escherichia coli* represents a critical public health threat, yet their genomic characteristics in community settings remain poorly described. We sequenced 77 multidrug-resistant isolates from children (n = 59), livestock (n = 17), and food (n = 1) in peri-urban Nairobi, Kenya. Phylogenetic analysis revealed polyphyletic diversity across phylogroups and sequence types without host-specific clustering. We detected high-risk lineages ST69 (n = 5) and ST131 (n = 2) among children. Nearly all isolates carried extended-spectrum β-lactamase genes, including $bla_{CTX-M-15}$ and $bla_{OXA-1}$, with resistance spanning nine antibiotic classes. Network analysis revealed a stable multidrug-resistance cluster ($bla_{TEM-1B}$, *aph(3)-Ib*, *aph(6)-Id*, *sul2*, *tetA*) shared across hosts. Virulence-associated gene profiling showed 34 enteric-associated determinants, with children's isolates carrying significantly more genes than livestock (mean 6.4 vs. 4.2, p = 0.001). The presence of virulent, multidrug-resistant lineages in apparently healthy community carriers highlights a potential reservoir of multidrug-resistant diarrheagenic-associated pathogenic potential outside hospitals. These findings underscore urgent need for genomic surveillance, stewardship and WASH to interrupt transmission of high-risk *E. coli* clones.

## Introduction

In a recent global analysis of AMR burden across 23 pathogens, 88 pathogen–drug combinations, and 204 countries, *Escherichia coli* ranked as the leading cause of AMR-associated mortality, with an estimated 829,000 AMR-associated deaths and 219,000 AMR-attributable deaths [1]. Whilst a gut commensal, *E. coli* includes pathotypes that cause invasive disease—including diarrhoea, urinary tract infection, and sepsis—and readily acquires resistance via mobile genetic elements. Children

**Data availability statement:** Raw sequencing reads generated in this study have been deposited in the NCBI Sequence Read Archive (SRA) under BioProject accession PRJNA1337292, publicly accessible.

**Funding:** The project Development of a comprehensive intervention to address foodborne enteric disease risks among young children living in low-income informal neighborhoods of Maputo and Nairobi This work was supported by the Bill & Melinda Gates Foundation (BMGF) and the Foreign, Commonwealth and Development Office (FCDO) of the UK Government (grant number INV-008449 to OC) and by the CGIAR Research Program on Agriculture for Nutrition and Health, led by the International Food Policy Research Institute (IFPRI). Additional support was provided by the Royal Society of Tropical Medicine and Hygiene (RSTMH) in partnership with Journal of Comparative Pathology Educational Trust (JCPET) through an early career research grant (grant to NOO). The funders had no role in study design, data collection and analysis, decision to publish, or preparation of the manuscript. No authors received a salary from any of the funders for this work.

**Competing interests:** The authors have declared that no competing interests exist.

under five are of particular concern, with multidrug-resistant pathogens responsible for about one in five AMR-attributable deaths in this age group, largely secondary to previously treatable infections [1].

Peri-urban settlements in low-resource settings—such as Dagoretti in Nairobi, Kenya—create optimal conditions for AMR emergence and transmission. Children in these environments are routinely exposed to bacterial pathogens due to poor sanitation, dense human–livestock cohabitation, informal and poorly regulated food chains, high antibiotic accessibility and misuse, immune deficiencies, and fragmented healthcare access [2,3]. Data are essential to quantify this problem, yet they remain scarce in community settings in low-resource contexts; most surveillance is hospital-based or focuses on adults. In our previous work, we showed that apparently healthy children aged 6–24 months carried multiple *E. coli* pathotypes, associated with younger age groups and specific foods [4]. Here, we extend this by analysing the population structure and antimicrobial resistance profiles of multidrug-resistant *E. coli* isolated from children, cohabiting livestock, and food sources in a rapidly urbanising peri-urban community. Our findings provide actionable, community-level evidence to guide antimicrobial stewardship and One Health interventions, that can be targeted through integrated surveillance, tailored prescribing guidelines, focused hygiene and food-safety campaigns, and cross-sector policy reforms.

## Methods

### Ethics statement

The study was approved by the Research Ethics Committee of the London School of Hygiene and Tropical Medicine (Ref: 17188) and the Institutional Research Ethics Committee at the International Livestock Research Institute (Ref: ILRI-IREC2019–26). Livestock sampling was approved by the ILRI Institute of Animal Care and Use Committee (Ref: ILRI-IACUC2020-15). All study participants (i.e., adult caregivers, food vendors, and livestock owners) provided written informed consent before entry into the study.

### Study design, microbiology and antimicrobial susceptibility testing

We conducted a cross-sectional study of 585 households with at least one child aged 6–24 months between May 1 and November 30, 2021. Sampling and microbiological methods are detailed elsewhere [4] and in the Supplementary Methods (S1 Text). Briefly, stool samples were collected from children (6–24 months), food samples, and livestock faecal samples (cattle, goats, sheep, poultry, and pigs). Samples were processed at the ILRI microbiology laboratories using standard procedures. Stool samples were directly plated on MAC and CT-SMAC agar; food samples were pre-enriched in BPW before plating on MAC and CT-SMAC. Presumptive *E. coli* colonies were tested for indole production using motility–indole–ornithine media, and indole-positive colonies were confirmed by MALDI-TOF MS. Diarrhoeagenic *E. coli* (DEC) pathotypes were assigned based on the detection of canonical marker genes using pathotype-specific primers as described previously [4] and in the

Supplementary Methods. Antimicrobial susceptibility testing was performed using the Kirby–Bauer disk diffusion method in accordance with CLSI guidelines (CLSI breakpoints for *Enterobacterales* - Table 2A, 32$^{nd}$ Edition). Multidrug-resistant (MDR) *E. coli* isolates (defined as resistance to at least one antibiotic in three or more antibiotic classes) were then selected for whole-genome sequencing (WGS) and downstream analyses.

Among 973 *E. coli* isolates from stool/faeces and food (503 children, 282 livestock, 188 food), 274 diarrhoeagenic pathotypes underwent antimicrobial susceptibility testing (127 child, 122 livestock, 25 food). Of these, 82 isolates were classified as MDR and sequenced (59 child, 17 livestock, 6 food), originating from 75 of the 585 households (Fig 1). For most samples, a single *E. coli* isolate was selected for WGS. However, in six child samples, two distinct *E. coli* pathotypes were identified. Because both isolates from each of these samples exhibited multidrug resistance, both were selected for sequencing, resulting in a total of 12 isolates from six samples. These isolates were treated as independent genomes for downstream analyses.

## Whole genome sequencing and bioinformatic analyses

DNA was extracted with the TANBead Gram Bacteria DNA Auto Plate kit, and whole-genome sequencing was done at the International Livestock Research Institute on an Illumina NextSeq 550 (San Diego, CA, USA). The quality of raw reads was checked using FastQC v0.12.0 [5]. Poor-quality reads and sequence adaptors were trimmed using fastp v0.22.0. The quality score for qualified bases was 25 and the head-tail trimming mean-quality was 20 per sliding window of 4. The pre-processed reads were also filtered to only retain those with at least 20 bp. Sequencing data have been deposited in the NCBI Sequence Read Archive (SRA) under BioProject accession PRJNA1337292. Individual isolates accession numbers are provided in the supplementary material (S1 Table). Multilocus sequence types (MLST) were assigned using the Achtman scheme, and Clermont phylogroups were determined with the ClermonTyping tool v1.4.0 [6].

*De novo* assembly was done using SPAdes v3.15 using default settings [7] and the quality of the assembled genomes was evaluated using QUAST v5.0.2 [8]. QUAST quality thresholds applied were: (1) a minimum of 3Mb aligned to EC958 reference strain; (2) an assembly length not exceeding 6.5Mb; (3) GC content falling within the range of 50% to 51% (a descriptive QC metric to confirm species consistency; no assemblies were excluded solely on the basis of GC content); and (4) an assembly N50 greater than 30kb. Of the 82 genomes, five (originating from food samples) exceeded 6.5 Mb and were excluded from downstream analyses. A total of 77 genomes were included in the downstream analyses.

Using EC958 (Accession: GCA_000285655.3) as the reference genome, the pre-processed reads were mapped and core genome alignment generated using snippy v4.6.0 [9]. Core SNPs were extracted and the core alignment visualized on seaview. Gaps were removed using bmge [10] with the options: -t DNA -g 0.10 -b 5 -h 0:1 -of). Gubbins was used to identify and mask putative recombinant regions and a maximum-likelihood phylogeny was built from the recombination-free SNP alignment using IQ-TREE v2.2.0 [11], with ModelFinder for model selection and ultrafast bootstrap approximation (1,000 replicates) for branch support [11,12]. The tree was visualized and annotated in R with ggtree, incorporating metadata (source, sequence type, ST complex, phylogroup).

## In silico identification of AMR determinants and plasmid replicons

Acquired antibiotic resistance genes were identified from assemblies with starAMR (≥95% identity, ≥60% coverage) against the ResFinder database. Chromosomal fluoroquinolone resistance was assessed using PointFinder to detect amino acid substitutions in the quinolone resistance–determining regions of *gyrA* and *parC*. Plasmid replicons were predicted with PlasmidFinder and virulence-associated genes predicted using VirulenceFinder database (≥90% identity, ≥60% coverage).

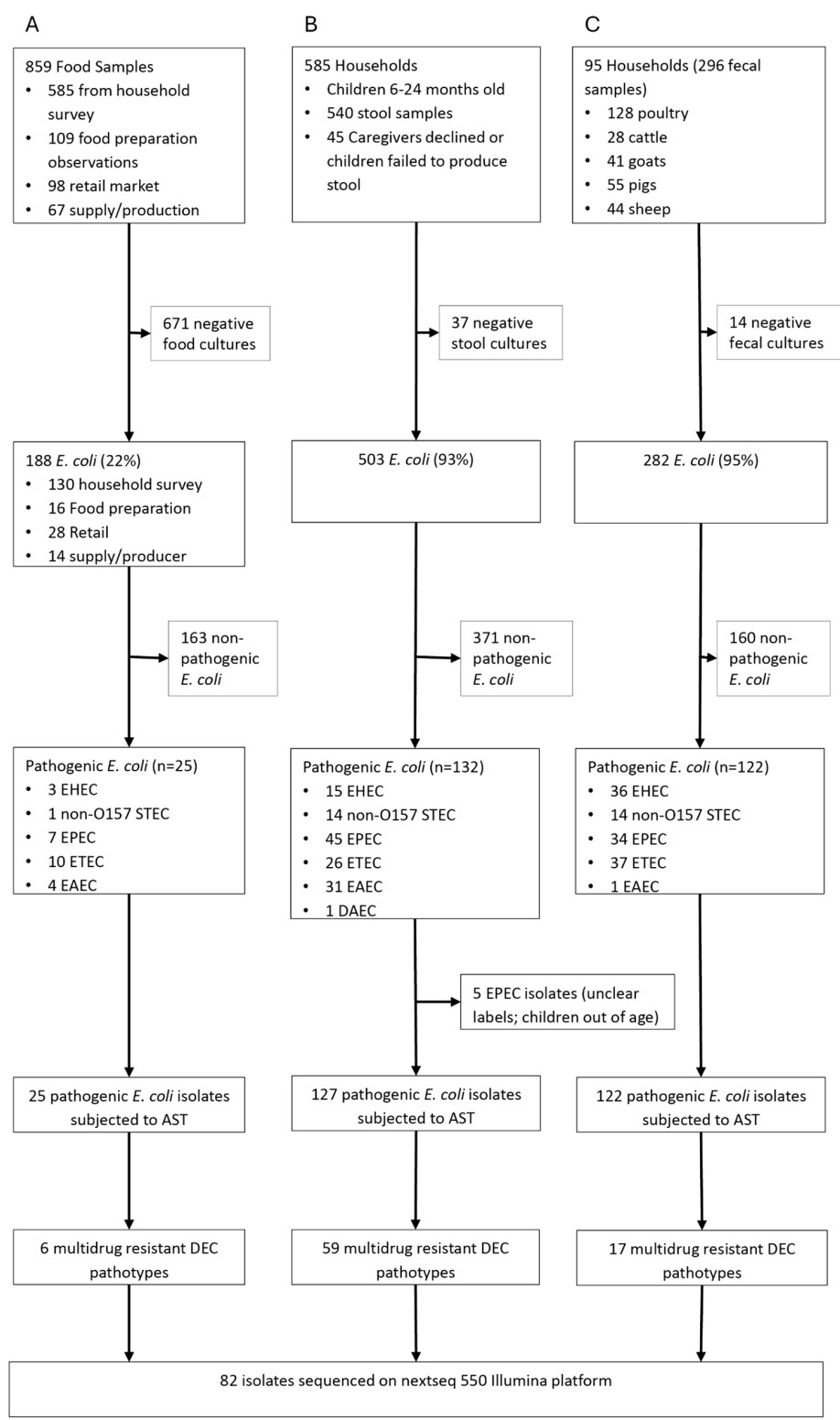

**Fig 1. Flowchart showing the number of isolates from (A) Food samples, (B) Children stool samples and (C) livestock faecal samples subjected to genotypic analysis by whole genome sequencing.** DEC is diarrheagenic *E. coli*.

## Statistical analysis

We first compared the distribution of ARGs and plasmid replicons across source types using the Kruskal–Wallis test (with post-hoc Dunn corrections where appropriate). Community composition was then evaluated with PERMANOVA on Bray–Curtis dissimilarities with the adonis2 function in the Vegan package in R. Both the p-value and $R^2$ (proportion of variance explained by source type) were reported. Ordination via non-metric multidimensional scaling (NMDS) was used to visualise clustering and dispersion among sources.

## Network analysis

We used a network-based approach to investigate patterns of co-occurrence among acquired ARGs across all isolates. A co-occurrence network was constructed using binary presence-absence data for each gene. Pairwise Jaccard similarity was calculated to assess the degree of co-occurrence between gene pairs, and statistically significant associations were identified using the disparity filter (α = 0.05) implemented in the backbone R package. The resulting backbone network was analysed for community structure using the Louvain algorithm to identify clusters of ARGs (modules) and visualized with the igraph and ggraph packages to highlight dominant multidrug resistance gene clusters. Network edges were defined based on pairwise co-occurrence frequencies of resistance genes across isolates; no contig-level analysis or plasmid reconstruction was performed.

## Results

### Diarrheagenic *E. coli* isolates exhibit a highly diverse population structure

A total of 77 *E. coli* genomes were analysed: 59 (76.6%) from children, 17 (22.1%) from livestock, and 1 (1.3%) from food. These isolates were distributed across 70 households. Genomes spanned all major phylogroups including cryptic clade I: A (31·2%), B1 (26·0%), D (19·5%), B2 (13·0%), E (2·6%), F (1·3%), G (1·3%), and clade I (1·3%). Phylogroup A predominated in livestock (58·8% of livestock isolates), whereas B1 was most common among children (32·2%). Overall, 46 known sequence types (STs) and 6 unknown STs based on the Achtman seven-locus scheme were identified with the most frequent being ST10 (7·8%), ST69 (6·5%), and ST31 (6·5%). Only two STs (ST10 and ST48) were shared between children and livestock isolates, while all other STs were exclusive to one host group. Two globally recognised high-risk *E. coli* lineages were detected among the children's isolates: ST69 (n = 5), assigned to phylogroup D, and ST131 (n = 2), assigned to phylogroup B2. Grouping isolates into clonal complexes showed CC10 was most common (16/77, 23·4%), occurring in 29·4% of livestock isolates and 18·6% of child isolates. We generated a core-genome alignment of 94,928 conserved nucleotide positions across all 77 genomes to infer their phylogenetic relationships (Fig 2). One genome belonging to *Escherichia* cryptic Clade I showed substantial genetic divergence from the *E. coli* sensu stricto lineages and was excluded from the final phylogenomic analysis to preserve resolution of the core *E. coli* population structure; this genome is shown in S1 Fig. Our findings revealed that genomes distributed throughout the whole phylogenetic tree, suggesting that clustering occurred by phylogroup rather than by host type (Fig 2). Human, livestock, and food isolates were interspersed within the same phylogroups, and in several cases, isolates from different hosts shared the same STs. However, within these shared STs, genomes often exhibited genetic diversity, suggesting that while human and animal isolates can belong to the same ST, they are not always closely related at the core-genome level. This highlights both the overlap in population structure across hosts and the genomic heterogeneity present even within individual STs.

### Diarrheagenic *E. coli* isolates harbour diverse and abundant antibiotic resistance genes

By study design, all isolates selected for inclusion were phenotypically resistant to at least three antibiotic classes [13]. We identified 50 ARGs, along with 8 point mutations (four in *parC* region, three *gyrA* and one *parE*) (Fig 3). In total, these genes and mutations confer resistance to nine antibiotic classes. On average, isolates harboured 7.8 ARGs (range 2–22).

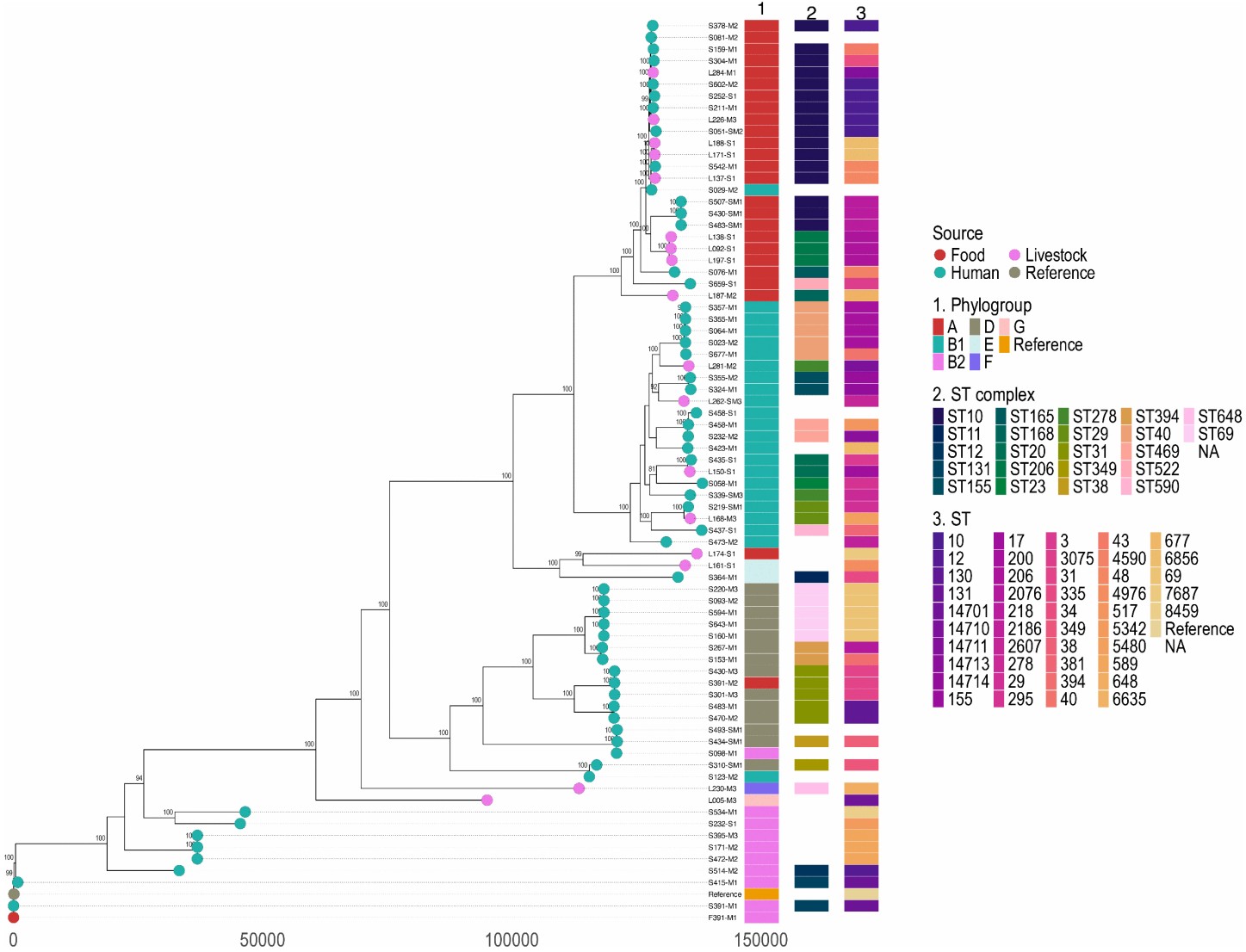

**Fig 2. Maximum-likelihood phylogenetic tree generated from recombination-masked single nucleotide polymorphism alignment of 77 *E. coli* genomes obtained from children, food and livestock in Dagoretti South subcounty, Nairobi Kenya.** Values at nodes represent Ultrafast Bootstrap support (only values ≥80% are shown). The scale bar represents the genetic distance in number of single nucleotide polymorphisms. The tree tips are annotated with isolates source. Further annotation is done by Sequence Type (ST), phylogroups, ST complex.

The most common ARGs were $bla_{TEM-1B}$ (83.1%), *sul2* (77.9%), *aph(6)-Id* (76.6%), *aph(3)-Ib* (74.0%), and *tetA* (63.6%). Notably, 72 isolates (93.5%) carried at least one of nine extended-spectrum β-lactamase (ESBL) genes—96.6% of isolates from children, 82.4% from livestock, and the single food isolate. Six children's isolates and the food isolate carried $bla_{CTX-M-15}$ gene, and four human isolates plus the food isolate harboured $bla_{OXA-1}$. The single food isolate was pan-resistant, carrying 23 AMR genes spanning all antibiotic classes except fosfomycin. The distributions of most (52/58, 89.7%) AMR genes and mutations did not differ significantly between children and livestock. Nevertheless, five ARGs (*aadA1*, *aadA2*, *cmlA1*, *sul3*, *tetA*) were significantly more common in livestock isolates, whereas *dfrA8* was common in children (Fisher's exact test, Benjamini–Hochberg corrected $p < 0.05$).

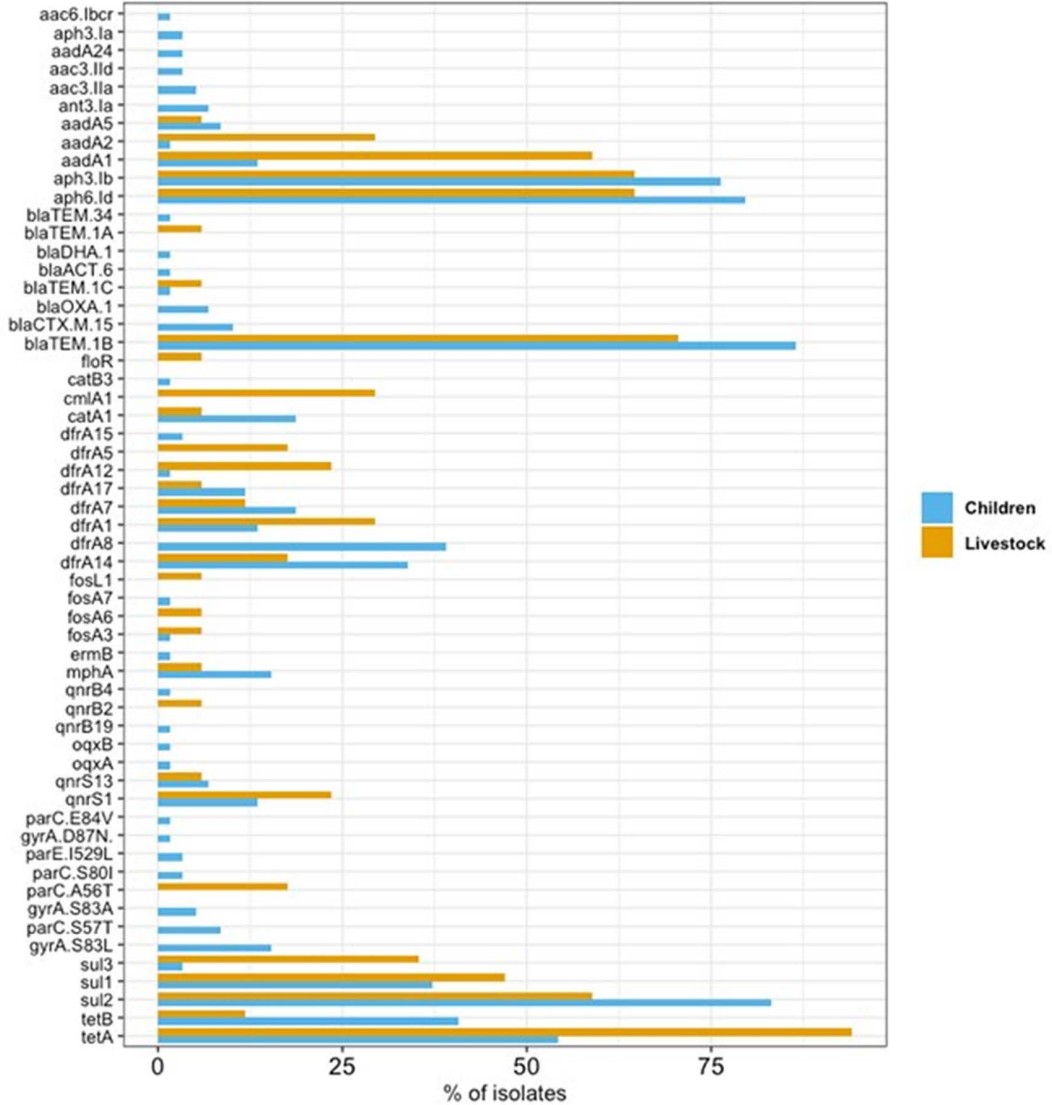

**Fig 3. Percentage of *E. coli* isolates from 59 Children (blue) and 17 Livestock (orange) carrying each AMR mechanism.** Percentages are calculated within each source group.

The number of acquired ARGs (excluding point mutations) did not differ significantly between children and livestock isolates (median 7 vs. 8; $p > 0.05$, Fisher's test), although the small number of livestock isolates may limit statistical power ([Fig 4]). Similarly, PERMANOVA showed no significant difference in ARG community composition between sources ($R^2 = 0.02$, $p > 0.05$), indicating that source type explains only ~2% of the variation in AMR gene profiles and that there is substantial overlap in AMR gene composition among *E. coli* isolates from children and livestock. Principal coordinates analysis (PCoA) showed substantial overlap among isolates from different sources, with the two axes explaining 22.8% (PCoA1) and 15.0% (PCoA2) of the total variance.

PLOS Global Public Health

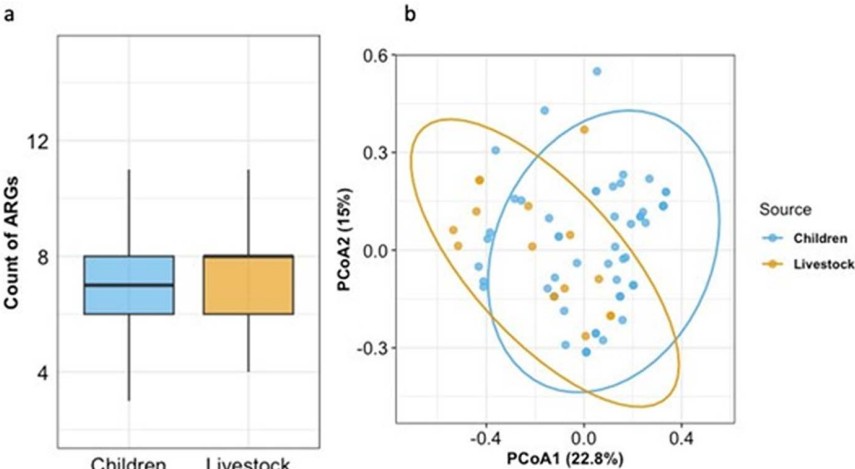

**Fig 4. Distribution and composition of acquired ARGs in Escherichia coli from children and livestock.** (a) Boxplots of acquired ARG counts per isolate (b) PCoA of Jaccard distances based on acquired ARG presence/absence in E. coli isolates from Children (blue) and Livestock (orange). Points show individual isolates; ellipses denote 95% confidence regions.

### Network analysis reveals a central multidrug resistance cluster

Network analysis of acquired ARGs identified a densely connected cluster consisting of five ARGs ($bla_{TEM-1B}$, aph3-Ib, aph6-Id, sul2 and tetA), with statistically significant co-occurrence relationships detected using a disparity filter (α = 0.05). This core multidrug resistance cluster was detected in 29 isolates (37.7% of all samples), including 22 isolates from children (37.3%), 6 from livestock (35.3%), and the single food isolate (Fig 5). The MDR cluster represents statistical co-occurrence of resistance determinants across isolates and does not imply physical genetic linkage, plasmid carriage, or co-localization on the same mobile element.

### Genes encoding virulence factors associated with diarrheagenic *E. coli*

We identified 194 virulence-associated genes and compiled a bespoke list of 34 virulence-associated factors that have been previously described in established diarrheagenic and extraintestinal *E. coli* pathotypes. These genes have been reported in *E. coli* lineages historically associated with intestinal and extraintestinal infections in humans [14]. However, their detection in this study reflects gene carriage rather than confirmed pathogenicity, virulence expression, or clinical outcome. The most prevalent gene was alpha hemolysin toxin E *(hlyE*, 84.4%, 65/77 isolates), followed by colonisation-associated type 1 fimbriae (*fimH*, 80.5%) and colonisation-associated curli fimbriae *csgA* (58.4%). Of the 34 genes, ten were associated with colonization (*tir, nleB, csgA, fimH, eae, toxB, afaA, aggR, aap, aafA*), eleven with effectors (*espB, espF, espA, nleA, cif, espJ, espC, espP, etpD, ibeA, aaiC*), nine with toxins (*hlyE, sat, astA, elt, pic, sigA, sepA, pet, eatA*), and four with fitness (*fyuA, iutA, kpsE, kpsMII*) (S2 Table). Isolates from children carried significantly higher virulence scores compared with those from livestock (mean 6.4 vs. 4.2; *p* = 0.001, Mann-Whitney U). Across phylogroups, B1 and B2 isolates exhibited the highest virulence scores (mean 7.4 and 7.3, respectively), whereas isolates from phylogroup E (mean 6.5) and A (mean 4.8) carried fewer virulence determinants. We investigated virulence gene carriage in more detail among the high-risk lineages ST131 (n = 2) and ST69 (n = 5). All seven isolates harboured the fitness associated ferric receptor gene (*fyuA*), the colonisation-associated type 1 fimbrial adhesin gene (*fimH*), and the fitness-associated aerobactin receptor gene (*iutA*), while the alpha hemolysin toxin gene (*hlyE*) was present in all ST69 isolates.

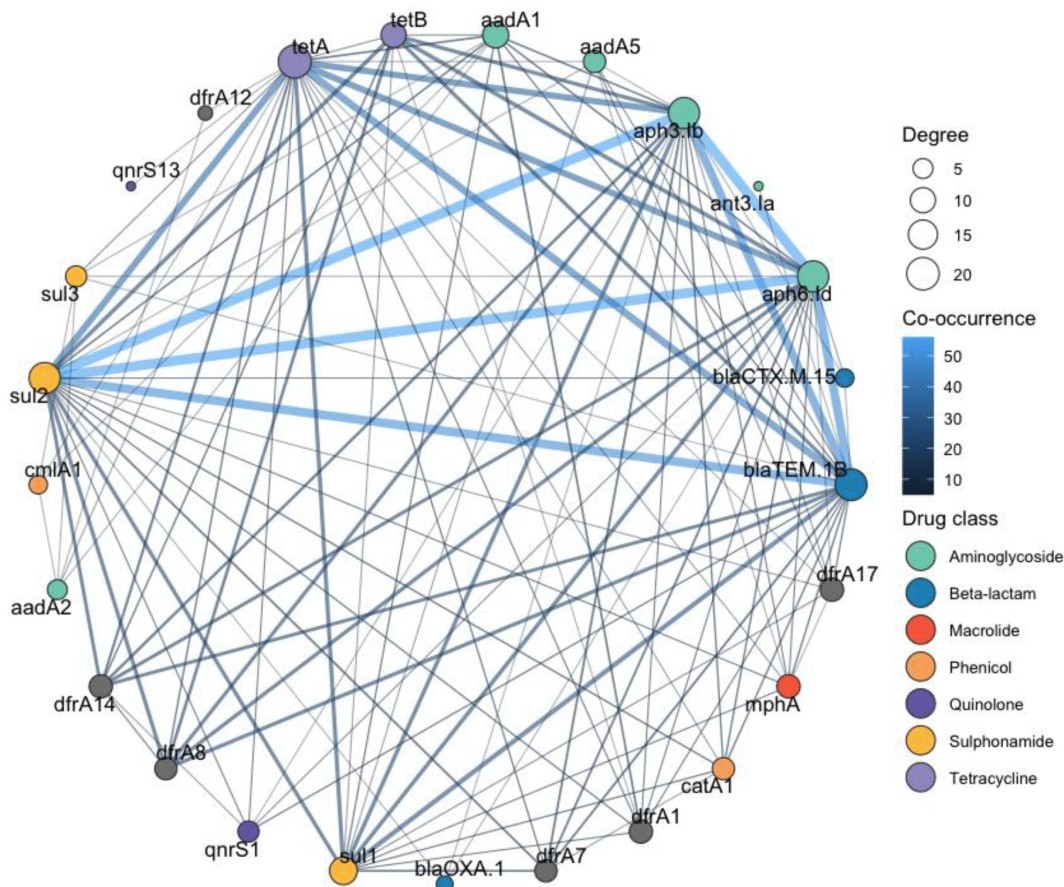

**Fig 5. Co-occurrence network of acquired antimicrobial resistance genes (ARGs).** Nodes represent individual ARGs; edges indicate significant co-occurrence relationships (disparity filter, α = 0.05). Node size and edge thickness reflect frequency and co-occurrence strength.

## Discussion

Our genomic analysis shows that diarrheagenic *E. coli* circulating in children, livestock, and food in a peri-urban setting of Nairobi exhibit a highly diverse population structure, extensive AMR gene carriage, and broad virulence potential. Together, these findings underscore the complexity of AMR *E. coli* population structure at One Health interfaces in low-income settings, where intense microbial exposures, close human–livestock contact, and unregulated antimicrobial use drive the amplification and persistence of multidrug resistance and virulence.

We detected all major *E. coli* phylogroups, including cryptic clade I, alongside 46 distinct sequence types indicating bacterial population heterogeneity and absence of host-restricted structure. This mirrors previous observations from sub-Saharan Africa and elsewhere that commensal and pathogenic *E. coli* exhibit broad phylogenetic diversity across hosts, with limited evidence of strict host adaptation [2,15–18]. The absence of phylogenetic clustering by host type in our core-genome phylogeny further supports widespread genetic mixing, suggesting overlapping reservoirs and/or acquisition from shared environmental sources such as water, soil, or food handling chains [19,20]. Importantly, our finding of globally disseminated, clinically relevant lineages (ST69 and ST131) from children highlights the capacity of high-risk pathogenic and resistant clones to circulate endemically outside healthcare settings, thereby reinforcing their role as archetypal "community pathogens" [21,22].

By design, our study targeted multidrug-resistant isolates, yet the breadth of AMR genes identified, spanning nine antibiotic classes, was striking. Nearly all isolates (93.5%) carried at least one of nine ESBL genes, with $bla_{CTX-M-15}$ and $bla_{OXA-1}$ detected in a subset of human and food isolates. The detection of these clinically significant genes in apparently healthy children under five years of age has major public health implications. It raises critical questions about the likely sources of such resistance determinants, which are traditionally associated with hospital environments. They may reflect substantial selection pressures from community-level antibiotic use, transmission within households (including via caregivers and food), or circulation through broader contaminated environments that receive both human and animal waste. Although our study was not designed to disentangle these pathways, clarifying their relative contributions should be a priority for future research.

In Nairobi and across Kenya, antibiotic use in children under five is high and often poorly regulated, creating strong selection pressure for resistance. In one informal settlement, children accounted for nearly two-thirds of household antibiotic use, with amoxicillin, ampicillin, and cotrimoxazole most common [23]. During the COVID-19 pandemic, 97% of outpatient and community pharmacy visits for respiratory infections led to antibiotic prescriptions, with young children disproportionately receiving WHO "watch-group" drugs, underscoring community drivers of ESBL persistence [24].

The substantial overlap of resistome profiles between children and livestock isolates, mirroring recent genomic surveys that demonstrate shared AMR communities across human, animal, and water [25], underscores the permeability of ecological boundaries and ARG flow across One Health interfaces or overlapping selection pressures. We found a densely connected MDR cluster consisting of $bla_{TEM-1B}$, *aph(3)-Ib*, *aph(6)-Id*, *sul2*, and *tetA*, present across more than one third of isolates from all sources. This cluster has been identified in prior studies [26] across phylogenetically diverse isolates underscoring its potential role as an ecological "backbone" of resistance in the community where it supports a stable and transmissible reservoir that may underpin the persistence of resistance in the community despite variable antimicrobial exposures [26–29]. This co-occurrence might indicate the involvement of mobile genetic elements, likely conjugative plasmids, capable of co-transferring multiple resistance determinants in a single event, thereby facilitating the rapid acquisition of a multidrug-resistant phenotype and posing substantial public health concern. Alternatively, this may reflect overlapping or cumulative antibiotic exposures at the individual and community level that sustain selection pressures. Future work using long read sequencing and plasmid reconstruction will be critical to unravel the genetic vehicles sustaining this backbone, clarify its role in horizontal gene transfer across One Health compartments, and identify points of intervention to disrupt its transmission.

We show that *E. coli* isolates from apparently healthy children and livestock display a high diversity of resistance mechanisms and virulence-associated gene profiles, which represent a possible significant concern for public health and food safety [30]. These genes, central to colonisation, persistence, and host-cell damage, are maintained even in the absence of overt disease. Their co-occurrence in isolates from healthy children suggests that asymptomatic carriage constitutes a silent but functionally equipped reservoir capable of persistence, dissemination, and opportunistic transition to pathogenic lifestyles under favourable host or ecological conditions. Importantly, community carriage contributes to a substantial infection burden, as asymptomatic individuals can develop diarrheal disease or severe extraintestinal infections, amplifying transmission and adverse clinical outcomes [31,32]. The emergence of hypervirulent *E. coli* pathotypes in community and non-human reservoirs, exemplified by the fatal EAEC/STEC O104:H4 outbreak in Europe linked to sprouts [33,34], underscores the urgency of One Health surveillance to detect and contain such strains [35].

Future research should explicitly interrogate the epidemiological linkage between the genomic profiles of community-associated *E. coli* and the risk of clinical disease. While our findings reveal extensive carriage of virulence and resistance determinants in asymptomatic children, we were unable to establish whether these genomic signatures translate into increased risk of infection or adverse outcomes. Adequately powered longitudinal cohort studies and case-control designs integrating genomic, clinical, and epidemiological data will be critical for disentangling the functional relevance of specific virulence-resistance traits within the community.

A key limitation of this study is that whole-genome sequencing was performed exclusively on multidrug-resistant diarrhoeagenic *E. coli* isolates. Consequently, the population structure, resistance burden, and virulence profiles described here should not be interpreted as representative of the general community *E. coli* population, but rather of MDR DEC specifically.

## Conclusion

In conclusion, diarrheagenic *E. coli* circulating among apparently healthy children and livestock in Nairobi are genetically diverse, frequently multidrug-resistant, and carry multiple virulence-associated genes, including globally important high-risk lineages. Their detection in community settings highlights the permeability of One Health boundaries and the existence of silent community reservoirs of AMR. Yet current Infection Prevention and Control (IPC) strategies remain predominantly hospital-focused, with little attention to community transmission. By demonstrating that apparently healthy children and livestock harbour virulent, multidrug-resistant *E. coli*, our study positions communities as overlooked reservoirs of infection risk. Addressing this gap requires pragmatic approaches. While community-based genomic surveillance would be invaluable, implementation in low-resource settings faces challenges of cost, infrastructure, and sustainability. In the context of finite resources, strengthening passive surveillance systems, improving antimicrobial stewardship at the community level, and integrating feasible interventions such as WASH improvements and targeted caregiver education may provide greater immediate returns. Active genomic surveillance could then be prioritized for sentinel sites or outbreak settings where the risk of spillover across One Health interfaces is highest.

## Supporting information

**S1 Table. Table showing individual isolate accession numbers in Sequence Read Archive- SRA.**
(TSV)

**S2 Table. Table showing the frequency of virulence factors from 77 *E. coli* genomes.**
(XLSX)

**S1 Text. Supplementary methods.**
(DOCX)

**S1 Fig. Maximum-likelihood phylogeny of 77 *E. coli* isolates.** The tree was constructed using IQ-TREE2 based on a recombination-masked SNP alignment. Values at nodes represent Ultrafast Bootstrap support (only values ≥80% are shown). The scale bar represents the genetic distance in number of single nucleotide polymorphisms.
(TIF)

## Author contributions

**Conceptualization:** Noah O. Okumu, Elizabeth A.J. Cook.

**Data curation:** Noah O. Okumu, John Juma, Dishon M. Muloi.

**Formal analysis:** Noah O. Okumu, John Juma, Kennedy Mwangi, Gilbert Kibet, Dishon M. Muloi.

**Funding acquisition:** Noah O. Okumu, Samuel Oyola, Arshnee Moodley, Oliver Cumming, Elizabeth A.J. Cook.

**Investigation:** Noah O. Okumu.

**Methodology:** Noah O. Okumu, John Juma, Linnet Ochieng, Dishon M. Muloi.

**Project administration:** Noah O. Okumu.

**Resources:** Samuel Oyola.

**Supervision:** Samuel Oyola, Arshnee Moodley, Joseph J.N. Ngeranwa, Oliver Cumming, Elizabeth A.J. Cook.

**Validation:** Noah O. Okumu, Dishon M. Muloi.

**Visualization:** Noah O. Okumu, Dishon M. Muloi.

**Writing – original draft:** Noah O. Okumu, Dishon M. Muloi.

**Writing – review & editing:** Noah O. Okumu, John Juma, Samuel Oyola, Arshnee Moodley, Kennedy Mwangi, Gilbert Kibet, Linnet Ochieng, Julie Watson, Joseph J.N. Ngeranwa, Oliver Cumming, Elizabeth A.J. Cook, Dishon M. Muloi.

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
