## [Decision Letter · Decision Letter 0]

27 Jan 2026

PGPH-D-25-03679

Genomic diversity of pathogenic multidrug-resistant Escherichia coli across asymptomatic children and livestock in Nairobi, Kenya

Dear Dr. Okumu,

Thank you for submitting your manuscript to PLOS Global Public Health. After careful consideration, we feel that it has merit but does not fully meet PLOS Global Public Health’s publication criteria as it currently stands. Therefore, we invite you to submit a revised version of the manuscript that addresses the points raised during the review process.

Please revise your manuscript according to the comments of both reviewers; in addition, please provide a point-by-point answer to the comments from each reviewer in the "Response to Reviewers" document. Indicate the corresponding edits in the revised manuscript (page and line number) in this "Response to Reviewers" document. Please also see below for detailed instructions.

We look forward to receiving your revised manuscript.

Kind regards,

Hui-min Neoh

Academic Editor

Journal Requirements:

i. Please clarify all sources of financial support for your study. List the grants, grant numbers, and organizations that funded your study, including funding received from your institution. Please note that suppliers of material support, including research materials, should be recognized in the Acknowledgements section rather than in the Financial Disclosure.

ii. State the initials, alongside each funding source, of each author to receive each grant. For example: "This work was supported by the National Institutes of Health (####### to AM; ###### to CJ) and the National Science Foundation (###### to AM)."

iii. State what role the funders took in the study. If the funders had no role in your study, please state: “The funders had no role in study design, data collection and analysis, decision to publish, or preparation of the manuscript.”

iv. If any authors received a salary from any of your funders, please state which authors and which funders.

2. Please ensure that your Ethics Statement is available in its entirety at the beginning of your Methods section, under a subheading 'Ethics Statement'.

3. Please upload separate figure files in .tif or .eps format. Also, remove the figures from your manuscript file but keep the legends.

4. We note that your Data Availability Statement is currently as follows: Data is contained within the article.

Additional Editor Comments (if provided):

Reviewers' comments:

Reviewer's Responses to Questions

**Comments to the Author**

1. Does this manuscript meet PLOS Global Public Health’s publication criteria? Is the manuscript technically sound, and do the data support the conclusions? The manuscript must describe methodologically and ethically rigorous research with conclusions that are appropriately drawn based on the data presented.? Is the manuscript technically sound, and do the data support the conclusions? The manuscript must describe methodologically and ethically rigorous research with conclusions that are appropriately drawn based on the data presented.

Reviewer #1: Yes

Reviewer #2: Yes

2. Has the statistical analysis been performed appropriately and rigorously?

Reviewer #1: Yes

Reviewer #2: No

3. Have the authors made all data underlying the findings in their manuscript fully available (please refer to the Data Availability Statement at the start of the manuscript PDF file)?

The PLOS Data policy requires authors to make all data underlying the findings described in their manuscript fully available without restriction, with rare exception. The data should be provided as part of the manuscript or its supporting information, or deposited to a public repository. For example, in addition to summary statistics, the data points behind means, medians and variance measures should be available. If there are restrictions on publicly sharing data—e.g. participant privacy or use of data from a third party—those must be specified.requires authors to make all data underlying the findings described in their manuscript fully available without restriction, with rare exception. The data should be provided as part of the manuscript or its supporting information, or deposited to a public repository. For example, in addition to summary statistics, the data points behind means, medians and variance measures should be available. If there are restrictions on publicly sharing data—e.g. participant privacy or use of data from a third party—those must be specified.

Reviewer #1: Yes

Reviewer #2: No

4. Is the manuscript presented in an intelligible fashion and written in standard English?

Reviewer #1: Yes

Reviewer #2: Yes

Reviewer #1: 1. Line 55, which food sample was this?

2. Line 62, specify which CLSI guidelines you used for ASTs.

3. Line 71, did the vendors give consent to have their behaviour and manners recorded and observed?

4. Line 100, what were the thresholds of identity and length for reproducibility?

5. Line 129, could you clarify what this statement, only two STs( ST10 and ST48) were found in both child and livestock.

6. Line 146, was there concordance between phenotypic and genotypic resistance patterns?

7. Line 156, were there specific ARGs found in each sample type?

8. Line 178, it would be better to represent the genes encoding virulence factors associated with pathogenic E.coli. you also left out the word WITH.

Reviewer #2: Title: Genomic diversity of pathogenic multidrug-resistant Escherichia coli across asymptomatic children and livestock in Nairobi, Kenya - PGPH-D-25-03679

Overall assessment

This is a well-executed genomic epidemiology study addressing an important One Health question in a high-risk peri-urban setting. The integration of WGS, resistome, virulome, and network analyses is a clear strength, and the manuscript is generally well written and contextualized. However, several conceptual and methodological issues limit the strength of causal inference, particularly regarding transmission, pathogenicity, and public health implications. Some claims are over-interpreted given the cross-sectional design and sampling framework, and additional clarification, sensitivity analyses, and reframing are required.

Major comments

1. Selection bias and representativeness of sequenced isolates

Only multidrug-resistant diarrhoeagenic E. coli were selected for sequencing. This creates a strong selection bias that limits interpretation of population structure, AMR prevalence, and virulence burden. Statements about “cryptic reservoirs”, and “community-level pathogenic potential” are not generalizable to community E. coli populations, but only to MDR DEC. I recommend the authors to explicitly acknowledge in the Abstract and Discussion that findings apply only to MDR DEC, not to commensal E. coli, and rephrase claims implying population-wide AMR burden

2. Inference of cross-host transmission is overstated

The manuscript frequently implies cross-host transmission based on shared STs, ARG profiles, and phylogenetic interspersion, but no SNP-distance thresholds or transmission analyses are presented. Shared STs or ARG clusters do not imply recent transmission and the authors themselves note genetic heterogeneity within STs, which weakens transmission claims. The authors should tone down on the language from “frequent cross-host transmission” to “overlapping reservoirs” and add a SNP-distance distribution for isolates sharing STs across hosts, or explicitly state that distances exceed plausible transmission thresholds.

3. Network analysis interpretation is biologically speculative

The MDR “backbone” cluster is interesting, but the manuscript implies plasmid-level stability and ecological centrality without direct plasmid reconstruction. Co-occurrence does not confirm physical linkage and the same genes may co-occur due to historical selection rather than active transmission. I recommend the authors to explicitly state that the network reflects statistical co-occurrence, not confirmed genetic linkage.

4. Food isolate interpretation is disproportionate

One food isolate is described as “pan-resistant” and repeatedly highlighted; n=1 does not permit inference and itt risks overstating food-borne transmission. The authors are advised not to use a single isolate to extrapolate food-chain risks and to consider this as an illustrative case and not evidence.

5. Definition and interpretation of “pathogenic” E. coli

The term “pathogenic E. coli” is used broadly and repeatedly, but many isolates carry individual virulence-associated genes rather than coherent pathotype-defining gene sets. Genes like fimH, csgA, and hlyE are common in commensal strains. Additionally, the VirulenceFinder detects gene presence, not expression or pathogenic potential. My recommendation to the authors is to clarify that results indicate “virulence-associated gene carriage”, not confirmed pathogenicity and to consider reporting complete pathotype profiles per isolate, not just aggregate virulence scores.

Minor comments:

1. Methods

• Clarify whether one isolate per sample or multiple colonies were sequenced.

• Justify QUAST thresholds (e.g., GC 50–51% is narrow and may exclude valid E. coli).

• State explicitly whether recombination was accounted for in phylogenetic inference.

2. Statistics

• PERMANOVA results should report R² values, not just p-values.

• Fisher’s exact tests across many ARGs risk low power with n=17 livestock isolates—acknowledge this.

3. Figures

• Fig 2 would benefit from:

o Scale bar in SNPs

o Clear bootstrap support threshold

• Fig 3: clarify whether percentages are within-source denominators.

4. Data availability

• “Data is contained within the article” is not given an accurate SRA deposition; PRJNA1337292 is non-existent on the SRA database. Update the Data Availability Statement to include accession numbers.

**Do you want your identity to be public for this peer review?** For information about this choice, including consent withdrawal, please see our Privacy Policy..

Reviewer #1: No

Reviewer #2: No

---

## [Editor Report · Decision Letter 1]

11 Mar 2026

Genomic diversity of diarrheagenic multidrug-resistant Escherichia coli across asymptomatic children and livestock in Nairobi, Kenya

PGPH-D-25-03679R1

Dear Okumu,

We are pleased to inform you that your manuscript 'Genomic diversity of diarrheagenic multidrug-resistant Escherichia coli across asymptomatic children and livestock in Nairobi, Kenya' has been provisionally accepted for publication in PLOS Global Public Health.

Best regards,

Hui-min Neoh

Academic Editor